# Increased Drop in Activity of Alkaline Phosphatase in Plasma from Patients with Endocarditis

**DOI:** 10.3390/ijms241411728

**Published:** 2023-07-21

**Authors:** Amila Kahrovic, Thomas Poschner, Anna Schober, Philipp Angleitner, Leila Alajbegovic, Martin Andreas, Doris Hutschala, Ruud Brands, Günther Laufer, Dominik Wiedemann

**Affiliations:** 1Department of Cardiac Surgery, Medical University of Vienna, 1090 Vienna, Austria; amila.kahrovic@meduniwien.ac.at (A.K.); thomas.poschner@meduniwien.ac.at (T.P.); n01636105@students.meduniwien.ac.at (A.S.); n11809354@students.meduniwien.ac.at (L.A.); martin.andreas@meduniwien.ac.at (M.A.); guenther.laufer@meduniwien.ac.at (G.L.); 2Division of Cardiac Thoracic Vascular Anaesthesia and Intensive Care Medicine, Medical University of Vienna, 1090 Vienna, Austria; doris.hutschala@meduniwien.ac.at; 3Alloksys Life Sciences BV, 6708 PW Wageningen, The Netherlands; r.brands1@uu.nl; 4IRAS Institute, University of Utrecht, 3584 Utrecht, The Netherlands

**Keywords:** alkaline phosphatase, cardiac surgery, cardio-pulmonary bypass, endocarditis, kidney

## Abstract

(1) Infective endocarditis is a severe inflammatory disease associated with substantial mortality and morbidity. Alkaline phosphatase (AP) levels have been shown to change significantly during sepsis. Additionally, we previously found that a higher initial AP drop after cardiac surgery is associated with unfavorable outcomes. Therefore, the course of AP after surgery for endocarditis is of special interest. (2) A total of 314 patients with active isolated left-sided infective endocarditis at the Department of Cardiac Surgery (Medical University of Vienna, Vienna, Austria) between 2009 and 2018 were enrolled in this retrospective analysis. Blood samples were analyzed at different time points (baseline, postoperative days 1–7, postoperative days 14 and 30). Patients were categorized according to relative alkaline phosphatase drop (≥30% vs. <30%). (3) A higher rate of postoperative renal replacement therapy with or without prior renal replacement therapy (7.4 vs. 21.8%; *p* = 0.001 and 6.7 vs. 15.6%; *p* = 0.015, respectively) and extracorporeal membrane oxygenation (2.2 vs. 19.0%; *p* = 0.000) was observed after a higher initial alkaline phosphatase drop. Short-term (30-day mortality 3.0 vs. 10.6%; *p* = 0.010) and long-term mortality (*p* = 0.008) were significantly impaired after a higher initial alkaline phosphatase drop. (4) The higher initial alkaline phosphatase drop was accompanied by impaired short- and long-term outcomes after cardiac surgery for endocarditis. Future risk assessment scores for cardiac surgery should consider alkaline phosphatase.

## 1. Introduction

Infective endocarditis (IE) is a rare but severe manifestation of valvular disease associated with significant morbidity and mortality [1]. The Duke criteria are used for diagnostic purposes [2]. Besides extended antimicrobial therapy, surgery should be performed in the case of heart failure, uncontrolled infection and in order to prevent embolism. If indicated, surgery should be performed as early as possible [3,4].

Transient bacteremia, as it may occur after dental, gynecologic, gastrointestinal, or urologic procedures or even after daily activities such as brushing teeth, may lead to the bacterial colonization of heart valves, resulting in sepsis, causing a spectrum of end-organ damage [5]. With up to two-thirds of patients developing acute kidney injury (AKI) in the setting of IE, the end-organ damage of the kidney is one of the most common [6]. Renal inflammation and resulting hypoxia are considered to be the cause [7]. The binding of pathogen-associated molecular patterns (PAMPs) such as lipopolysaccharide (LPS) to Toll-like receptor 4 leads to the further release of inflammatory mediators [8]. The pervasive alkaline phosphatase (AP) neutralizes LPS and converts proinflammatory molecules to anti-inflammatory ones (such as adenosine) [7,9,10]. (Severely) initially elevated AP levels have been considered a surrogate parameter for infection or bacteremia [11,12,13]. Consequently, AP has been considered a treatment target for sepsis patients in improving AKI or survival (in animal models) [7,14,15].

Additionally, to the inflammatory process caused by endocarditis, cardiac surgery induces some systemic inflammation due to the heart–lung machine (HLM). In up to 30%, a systemic inflammatory response syndrome (SIRS) occurs, associated with an elevated risk of end-organ injury [16,17,18]. No sole mechanism is responsible, but various factors are considered to be contributors [7,19,20,21,22]. An increased consumption of AP during cardiac surgery or extracorporeal membrane oxygenation is associated with a higher morbidity and mortality rate [23,24,25,26].

This study aims to evaluate the AP metabolism in the setting of an inflammatory state in IE patients undergoing surgery. To our knowledge, no study evaluating the course of AP following cardiac surgery for IE has been published previously.

## 2. Results

### 2.1. Patient Population and Cut-off Values

After reviewing the exclusion criteria, 314 patients were included in the retrospective analysis. According to an initial drop in AP of ≥30% (with an area under curve (AUC) 0.638; sensitivity 71%; specificity 53% for 1-year mortality) two groups were created: one with 179 patients with an AP drop ≥ 30% and another with 135 patients with an AP drop < 30%. The selection of cut-off values was guided by previous research findings [25,26]. In the following sections, the first value always refers to the lower initial AP drop cohort.

### 2.2. Preoperative Characteristics

The higher initial AP drop cohort was at a significantly higher surgical risk (EuroScore II 9.1 (15.9;3.5) vs. 16.1 (34.3;6.4) %; *p* = 0.000), had a significantly higher rate of preoperative dialysis (5.9 vs. 13.4%; *p* = 0.030) and prosthetic valve endocarditis (PVE) (20.0 vs. 40.2%; *p* = 0.000). All other patient characteristics were similar among the cohort members (Table 1).

### 2.3. Procedural Data

A significantly higher rate of re-sternotomy (21.5 vs. 40.8%; *p* = 0.000) and double valve replacement (8.9 vs. 22.9%; *p* = 0.001) were observed after an initial higher AP drop. Surgical times were prolonged in the initial higher AP cohort: total surgery time (240 (300;195) vs. 348 (455;255) min; *p* = 0.000), total cardiopulmonary bypass time (110 (155;90) vs. 162 (237;117) min; *p* = 0.000) and total aortic cross-clamp time (81 (113;62) vs. 111 (159;79) min; *p* = 0.000). Procedural data are illustrated in Table 2.

### 2.4. Laboratory Data

While AP values at baseline were significantly higher in the higher initial AP drop cohort (74 (96;62) vs. 93 (120;76) U/L; *p* = 0.000), there was no significant difference at 30 days (116 (141;89) vs. 128 (172;99) U/L; *p* = 0.089). Patients with the higher initial AP drop required a longer time to reach baseline values (days till baseline value surpassed: 4 (6;3) vs. 6 (7;4) days; *p* = 0.000; baseline within 3 days: 43.7 vs. 33.0%; *p* = 0.052; baseline within 5 days: 74.1 vs 60.3 %; *p* = 0.011). The postoperative course of AP dependent on the initial AP drop is depicted in Figure 1. On postoperative day (POD) 30, AP levels were significantly higher compared to baseline in both cohorts (*p* = 0.000 for both). Figure 2 illustrates the differences of AP levels at baseline to POD 1 and POD 30.

Although there was no statistically significant difference at baseline CRP values (3.9 (8.3;1.7) vs. 4.5 (11.7;1.5) mg/dL; *p* = 0.456), a significantly higher value after 30 days in the higher initial AP drop cohort was observed (2.1 (5.7;0.7) vs. 4.6 (10.6;1.8) mg/dL; *p* = 0.001). Laboratory values are provided in Table 3.

### 2.5. Adverse Events and Mortality

An overview of adverse events and mortality is available in Table 4. A higher rate of revision due to bleeding (7.4 vs. 15.6%; *p* = 0.027), postoperative renal replacement therapy with or without a previous one (7.4 vs 21.8 %; *p* = 0.001 and 6.7 vs. 15.6%; *p* = 0.015) and extracorporeal membrane oxygenation (2.2 vs. 19.0%; *p* = 0.000) were seen after a higher initial AP drop. While there was a strong trend toward a prolonged intubation (*p* = 0.054), a prolonged ICU stay was significantly more likely in the higher initial AP drop cohort (31.9 vs. 48.0%; *p* = 0.004).

Short-term mortality (30 day: 3.0 vs. 10.6%; *p* = 0.010; in-hospital: 5.9 vs. 17.3%; *p* = 0.002; 1 year: 14.1 vs. 25.7%; *p* = 0.012) and long-term mortality (*p* = 0.008, see Figure 3) were significantly higher in the higher initial AP drop cohort.

ECMO—extracorporeal membrane oxygenation; prolonged hospital stay defined as longer than 30 days; prolonged ICU (intensive care unit) stay defined as more than 7 days; prolonged intubation defined as reintubated, longer as 48 h or tracheostoma.

## 3. Discussion

In line with previous studies by Schaefer et al. (mitral valve patients with impaired left ventricular function) and Poschner et al. (post-cardiotomy VA-ECMO patients), we were able to demonstrate a relation between a higher initial AP drop and morbidity and mortality in patients undergoing cardiac surgery for left-sided IE [25,26]. Not only short-term (i.e., hospital mortality), but also long-term survival (over ten years) was significantly impaired following a higher initial AP drop (*p* = 0.008).

Our study found a significantly higher rate of ECMO implantation after a higher initial AP drop (2.2 vs. 19.0%; *p* = 0.000), presumably due to the longer operation times and the higher preoperative risk found in this cohort. In-hospital mortality after post-cardiotomy ECMO support is reported in up to two-thirds of patients [27,28]. The significantly worse outcome after a higher initial AP drop may thus be attributed to or at least influenced by the higher rate of ECMO implantation (30-day mortality 3.0 vs. 10.6%; *p* = 0.010; in-hospital mortality 5.9 vs. 17.3%; *p* = 0.002; 1-year mortality 14.1 vs. 25.7%; *p* = 0.012). However, as was already shown in a previous work, ECMO support did not (significantly) increase AP consumption, which is evident in reaching baseline values within five days in both studies [25].

Prosthetic valve endocarditis occurs in up to 20% of all endocarditis cases and is associated with a worse prognosis [29,30]. This finding could also be highlighted in our study cohort. Indeed, PVE was approximately twice as common in the higher initial AP drop cohort (40.2 vs. 20.0%; *p* = 0.000). Even though preoperative AP levels were within the normal range in both cohorts, AP levels were significantly higher in the higher initial AP drop cohort (see Table 3). Interestingly, baseline AP values, as well as baseline CRP values, showed no significant difference in PVE versus native IE (baseline AP in PVE 87 (109; 71) vs. native 84 (113; 68) U/L; *p* = 0.465; CRP in PVE 4.3 (12.6; 1.3) vs. native 4.2 (8.7; 1.8); *p* = 0.617). Hence, we may conclude that PVE seems to be not related to a more pertinent systemic inflammation and that the significantly impaired outcome was instead primarily due to the longer surgical times (total surgical time: PVE 389 (485;320) vs. native 245 (319;205) min; total cardiopulmonary bypass time PVE 192 (266;147) vs. native 118 (172;94) min and total aortic cross-clamp time PVE 121 (173;95) vs. native 82 (120;62) min all with a *p* = 0.000). As demonstrated by Doenst T et al., Nissinen J et al., and others, longer operation times are an independent predictor of worse outcomes [31,32]. An increased permeability in the gastrointestinal tract induced by HLM leads to a wash-in of endotoxins into the bloodstream, inducing the release of proinflammatory mediators [20,22]. AP-mediated dephosphorylation may convert a portion into anti-inflammatory substances [33]. This process leads to the consumption and subsequent removal of systemic alkaline phosphatase from circulation by Kupffer cell activity [23,34,35]. Thus, a prolonged duration of surgery is also likely to lead to a wider decrease in AP due to a prolonged exposure to endotoxins and associated mediators.

Isoforms of AP are present in the proximal tubule cells of the kidney [36]. Without a complete understanding of the role of AP in the kidney, animal studies have demonstrated increased levels of AP in the urine after ischemia/reperfusion injury and the administration of LPS, thus suggesting an injury to the proximal tubule brush border in these situations [37,38]. In the receiver operating characteristic (ROC) analysis in our cohort, the drop in alkaline phosphatase was associated with the need for postoperative dialysis with an AUC of 0.668. After an initial more significant drop, the need for postoperative dialysis was significantly more frequent (*p* = 0.001). Interestingly, the rate of postoperative dialysis was significantly higher with and without preoperative dialysis in the cohort with a higher initial AP drop. This suggests that patients with and without injured renal parenchyma are significantly more likely to have AKI due to SIRS (represented by the higher AP drop). Patients with AKI have significantly worse outcomes after cardiac surgery [39].

Zhang et al. highlighted the alkaline phosphatase on the surface membrane of neutrophils as a good tool for differentiating between SIRS patients with or without bacteremia [40]. Similarly, Kerner A. et al. and Tung C.B. et al. showed (severely) initially elevated AP levels as a surrogate parameter for bacteremia [11,12]. A total of 82.5% of the total cohort had positive blood cultures present, which were associated with a significantly elevated CRP level (CRP positive BC 4.9 (11.9;1.9) vs. negative BC 2.7 (5.7;0.8) mg/dL; *p* = 0.001), however, without any differences in AP (AP positive BC 86 (114;70) vs. negative BC 79 (106;64) U/L; *p* = 0.116). Hence, the findings from Kerner et al. and Tung C.B. et al. could not be confirmed in this study—likely due to the discrepancy in time between the diagnosis of endocarditis and the baseline AP values measured in our study.

An impaired barrier function of the gastrointestinal tract (GIT) leads to a wash-in of LPS. (Intestinal) AP possesses a variety of pathways to control intrinsic inflammation, including the regulation of the gut microbiome, the maintenance of barrier function (via tight junction proteins), as well as direct and indirect anti-inflammatory effects [41]. Most of the pathways above are Toll-like receptor 4 (TLR4)-dependent. TLR4 is activated via pathogen-associated molecular patterns (PAMPs) as expressed by bacteria (LPS) [42,43]. Yet, TLR4 activation also negatively affects tight junction proteins and barrier integrity [44,45]. The complete pathomechanism is not yet fully understood.

Hamarneh et al. demonstrated an impaired barrier function of the GIT in patients deprived of enteral feeding due to a loss of AP expression reversible to AP supplementation [46]. Given the critical state of IE patients, they often lack adequate enteral feeding. Hence, considering additionally the inflammatory state of endocarditis patients with the high rate of a need for postoperative dialysis (and the likely even higher rate of any AKI) and the further AP consumption during cardiac surgery, the supplementation of AP may be especially promising in IE patients [8]. Bovine intestinal AP was found to lead to a release of endogenous tissue non-specific AP (TNAP) that may contribute to the maintenance of the harmony of the human immune system [47]. Currently, clinical trials (e.g., APPIRED III—NCT03050476 on ClinicalTrials.gov) are evaluating the external administration of AP during (elective) cardiac surgery and are expected to reduce morbidity after cardiac surgery. Presbitero et al. already evaluated the response to external AP supplementation during surgery in a mathematical model validated with clinical data [48]. Based on this, a model may be developed that allows clinicians to predict the near-future outcome of their patients concerning adverse responses and enable them to take preventive actions. Future risk assessment scores, including alkaline phosphatase, are yet to be developed.

## 4. Materials and Methods

### 4.1. Patients

All patients who required valve surgery for isolated left-sided IE between January 2009 and October 2022 at the Department of Cardiac Surgery (Vienna General Hospital, Vienna, Austria) were included in this retrospective data analysis. The ethics committee of the Medical University of Vienna approved this retrospective study (2101/2022). Modified Duke criteria were used for IE diagnosis [49]. Exclusion criteria included IE involvement of the right side, surgical valve repair, homograft implantation and missing laboratory values at baseline or on POD 1. After reviewing the exclusion criteria, 314 patients were analyzed.

### 4.2. Laboratory Data

Blood samples were drawn at baseline and on each of the first seven PODs as well as on POD 14 (±3) and 30 (±5). Baseline was defined as a preoperative blood draw within a maximum of 7 days before surgery. The normal range of alkaline phosphatase is between 40 and 130 U/L and is determined as a routine parameter in our department. According to the (modified) 2011 IFCG (International Federation of Clinical Chemistry and Laboratory Medicine) guidelines, an enzyme kinetic photometric assay was performed to determine the plasma concentration of alkaline phosphatase [50]. ALP p-nitrophenyl phospshate (pnpp) is converted to phosphate and p-nitrophenol in the presence of magnesium and zinc ions. One unit is defined as the amount of enzymes causing the hydrolysis of one micromole of p-nitrophenyl phosphatase per minute at 37 °C. AP activity in U/L is calculated by multiplying the extinction value per minute by a corresponding factor.

### 4.3. Follow-Ups

To evaluate the individual postoperative outcome and adverse events, the local patient documentation system AKIM was used. Additionally, federal statistics (Statistics Austria, Vienna, Austria) were used to determine survival rates.

### 4.4. Statistical Analysis

Categorical variables are presented by numbers and percentages. For continuous variables, the mean ± standard deviation (SD) is reported if the data follow a normal distribution; otherwise, the median and interquartile range (IQR) are provided. The normal distribution assumption is tested using the Kolmogorov–Smirnov test. The patients were grouped according to the initial AP drop.
AP drop = 1 − AP POD 1/AP Baseline

Cut-offs were determined using an ROC analysis for 1-year mortality. Furthermore, the first day when alkaline phosphatase returned or exceeded baseline level was determined.

The Mann–Whitney-U-Test was used to assess differences between the continuous variables with non-normal distributions. For the comparison of categorical variables, the chi-squared test was applied. Differences between AP at baseline and AP POD 1 and AP POD 30, respectively, were determined using the Wilcoxon signed-rank test. The findings are visually presented using boxplots. Survival was visualized with Kaplan–Meier curves and the differences between the groups were tested using the log rank test. All analyses were performed using SPSS version 27.0 (IBM Corp, Armonk, NY, USA). A two-sided *p*-value of less than 0.05 was considered to indicate statistical significance.

## 5. Conclusions

A significantly higher morbidity and an impaired outcome after a higher initial drop of alkaline phosphatase at 30 days, 1-year and long-term survival was evident. Future risk assessment scores for cardiac surgery should consider alkaline phosphatase.

## Figures and Tables

**Figure 1 ijms-24-11728-f001:**
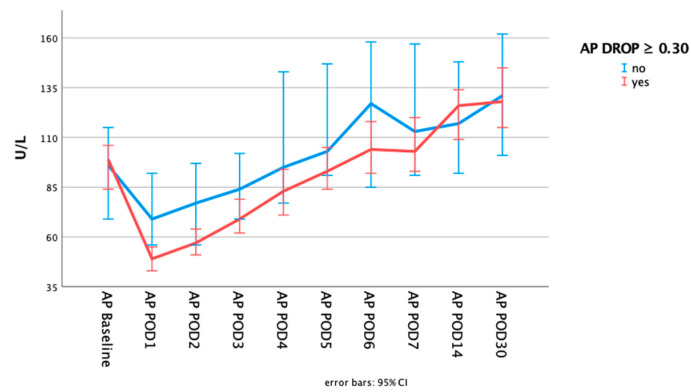
Alkaline phosphatase blood values dependent in the initial AP drop at baseline, on each consecutive day until POD 7 as well as on POD 14 and POD30.

**Figure 2 ijms-24-11728-f002:**
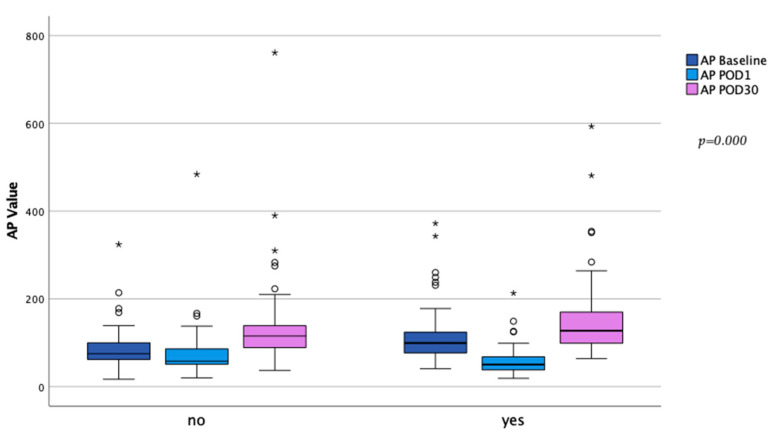
Comparison of alkaline phosphatase levels at baseline to POD 1 and POD 30 dependent on the initial drop of alkaline phosphatase. *p*-Values calculated for the difference in alkaline phosphatase between baseline to POD 1 and POD 30 using Wilcoxon signed-rank test (*p* = 0.000 for all). Circles highlight outliners (±1.5 × IQR), asterisk highlights extreme outliners (±3 × IQR).

**Figure 3 ijms-24-11728-f003:**
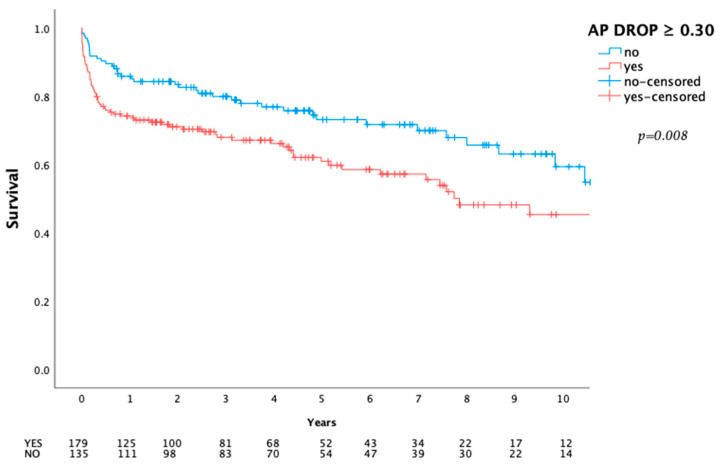
Ten-year survival. Long-term survival comparison dependent on initial drop of alkaline phosphatase. *p*-Value calculated using log-rank test.

**Table 1 ijms-24-11728-t001:** Preoperative characteristics.

		Overall n = 314	AP Drop < 30% n = 135	AP Drop ≥ 30% n = 179	*p*-Value ª
Age		62 (70;48)	59 (69;45)	62 (70;49)	0.230
Female		85 (27.1)	37 (27.4)	48 (26.8)	0.907
BMI		25.3 (28.9;22.8)	25.1 (28.9;22.9)	25.6 (28.7;22.5) ^†^	0.703
EuroScore II		11.7 (26.4;5.5)	9.1 (15.9;3.5)	16.1 (34.3;6.4)	0.000 *
NYHA III		78 (24.8)	34 (25.2)	44 (24.6)	0.902
NYHA IV		89 (28.3)	35 (25.9)	54 (30.2)	0.409
LVEF		60 (60;55)	60 (60;55)	60 (60;55)	0.042 *
Hypertension		200 (63.7)	79 (58.5)	121 (67.6)	0.098
Atrial fibrillation		90 (28.7)	38 (28.1)	52 (29.1)	0.861
IDDM		15 (4.8)	6 (4.4)	9 (5.0)	0.810
Preoperative dialysis		32 (10.2)	8 (5.9)	24 (13.4)	0.030 *
Cancer		19 (6.1)	10 (7.4)	9 (5.0)	0.381
h/o stroke		123 (39.2)	55 (40.7)	68 (38.0)	0.621
Modified Duke criteria					
Major criteria	Positive BC	259 (82.5)	107 (79.3)	152 (84.9)	0.192
	Vegetation	266 (84.7)	114 (84.4)	152 (84.9)	0.908
	Annular abscess	131 (41.7)	55 (40.7)	76 (42.5)	0.760
Minor criteria	IV drug abuse	24 (7.6)	11 (8.1)	13 (7.3)	0.770
	Fever > 38 °C	201 (64.0)	81 (60)	120 (67.0)	0.198
	Vascular phenomena	161 (51.3)	67 (49.6)	94 (52.5)	0.613
	Immunologic phenomena	19 (6.1)	11 (8.1)	8 (4.5)	0.176
PVE		99 (31.5)	27 (20.0)	72 (40.2)	0.000 *
Preoperative ventilation		142 (45.2)	64 (47.4)	78 (43.6)	0.499
Preoperative inotropic support		88 (28.0)	32 (23.7)	56 (31.3)	0.139
CPR		13 (4.1)	5 (3.7)	8 (4.5)	0.736
Lactate value		0.9 (1.3;0.7)	0.8 (1.2;0.7)	1.0 (1.3;0.7)	0.013 *

All values are referred in median (Q3; Q1) or in total number (n) and percentage (%) if not stated otherwise. |^†^ Normally distributed, median taken for better comparison. |ª If not stated otherwise, the Mann–Whitney *U* Test and Pearson’s chi-squared test, respectively, were used; values marked with an asterisk (*) achieved statistical significance. |age (years); BC—blood culture; BMI—body mass index (kg/m^2^); CPR—cardiopulmonary resuscitation; lactate (mmol/L); EuroSCORE II: European system for cardiac operative risk evaluation (%); h/o—history of; IDDM—insulin-dependent diabetes mellitus; IV—intravenous; NYHA—New York Heart Association classification; LVEF—left ventricular ejection fraction (%); PVE—prosthetic valve endocarditis.

**Table 2 ijms-24-11728-t002:** Procedural data.

	Overall n = 314	AP Drop < 30% n = 135	AP Drop ≥ 30% n = 179	*p*-Value ª
Urgent operation	262 (83.4)	118 (87.4)	144 (80.4)	0.100
Emergency operation	49 (15.6)	16 (11.9)	33 (18.4)	0.111
Salvage operation	3 (1.0)	1 (0.7)	2 (1.1)	0.734
Full sternotomy	303 (96.5)	128 (94.8)	175 (97.8)	0.159
Re-sternotomy	102 (32.5)	29 (21.5)	73 (40.8)	0.000 *
Isolated AVR	168 (53.5)	84 (62.2)	84 (46.9)	0.007 *
Isolated MVR	93 (29.6)	39 (28.9)	54 (30.2)	0.806
Double valve replacement	52 (16.9)	12 (8.9)	41 (22.9)	0.001 *
Surgery time	276 (390;220)	240 (300;195)	348 (455;255)	0.000 *
CPB	139 (204;104)	110 (155;90)	162 (237;117)	0.000 *
ACC	100 (144;70)	81 (113;62)	111 (159;79)	0.000 *

All values are referred in median (Q3; Q1) or in total number (n) and percentage (%). |ª If not stated otherwise, the Mann–Whitney *U* Test and Pearson’s chi-squared test, respectively, were used; values marked with an asterisk (*) achieved statistical significance. |ACC—aortic cross-clamp time in minutes; AVR—aortic valve replacement; CPB—cardiopulmonary bypass in minutes; MVR—mitral valve replacement.

**Table 3 ijms-24-11728-t003:** Laboratory data.

	Overall n = 314	AP Drop < 30% n = 135	AP Drop ≥ 30% n = 179	*p*-Value ª
Baseline AP value	85 (112;69)	74 (96;62)	93 (120;76)	0.000 *
Baseline CRP value	4.2 (10.7;1.6)	3.9 (8.3;1.7)	4.5 (11.7;1.5)	0.456
AP Drop	32.5 (45.2;23.0)	21.4 (25.8;11.5)	42.7 (54.7;36.4)	0.000 *
First day baseline AP value surpassed ^å^	5 (7;4)	4 (6;3)	6 (7;4)	0.000 *
Baseline within 3 days	118 (37.6)	59 (43.7)	59 (33.0)	0.052
Baseline within 5 days	208 (66.2)	100 (74.1)	108 (60.3)	0.011 *
30 day AP value ^∫^	121 (157;95)	116 (141;89)	128 (172;99)	0.089
30 day CRP value ^ç^	3.4 (7.8;1.1)	2.1 (5.7;0.7)	4.6 (10.6;1.8)	0.001 *

All values are referred in median (Q3; Q1) or in total number (n) and percentage (%). |ª If not stated otherwise, the Mann–Whitney U Test and Pearson’s chi-squared test, respectively, were used; values marked with an asterisk (*) achieved statistical significance. |AP in U/L |AP Drop—defined as 1—AP POD 1/AP Baseline. |CRP—C-reactive protein in mg/dL. |^å^ available for 254 patients (overall), 115 (AP drop < 30%) and 139 (AP drop ≥ 30%). ^∫^ Available for 140 patients (overall), 62 (AP drop < 30%) and 78 (AP drop ≥ 30%). ^ç^ Available for 159 patients (overall), 71 (AP drop < 30%) and 88 (AP drop ≥ 30%).

**Table 4 ijms-24-11728-t004:** Adverse events and mortality.

		Overall n = 314	AP Drop < 30% n = 135	AP Drop ≥ 30% n = 179	*p*-Value ª
Bleeding revision		38 (12.1)	10 (7.4)	28 (15.6)	0.027 *
Need for any renal replacement therapy		49 (15.6)	10 (7.4)	39 (21.8)	0.001 *
Need for any renal replacement therapy without preoperative		37 (11.8)	9 (6.7)	28 (15.6)	0.015 *
Need for ECMO		37 (11.8)	3 (2.2)	34 (19.0)	0.000 *
Prolonged intubation		107 (34.1)	38 (28.1)	69 (38.5)	0.054
Prolonged ICU stay		129 (41.1)	43 (31.9)	86 (48.0)	0.004 *
Prolonged hospital stay		62 (19.7)	22 (16.3)	40 (22.3)	0.182
Mortality					
	30 day	23 (7.3)	4 (3.0)	19 (10.6)	0.010 *
	In-hospital	39 (12.4)	8 (5.9)	31 (17.3)	0.002 *
	1 year	65 (20.7)	19 (14.1)	46 (25.7)	0.012*

All values are referred in median (Q3; Q1) or in total number (n) and percentage (%). |ª If not stated otherwise, the Mann–Whitney *U* Test and Pearson’s chi-squared test, respectively, were used; values marked with an asterisk (*) achieved statistically significance.

## Data Availability

The data presented in this study are available on request from the corresponding author. The data are not publicly available for reasons of data protection.

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
