# Peer review of "Increased Drop in Activity of Alkaline Phosphatase in Plasma from Patients with Endocarditis"

_ijms, 2023, doi:10.3390/ijms241411728_

Round 1

Reviewer 1 Report

Please change the title to „ Increased activity of alkaline phosphatase in plasma from patients with endocarditis”

Please seek statistical advice: have you tested with multivariate analysis?

On what basis did you chose as a cut off larger or less by 30 % (line 20)?

How do you define an infective endocarditis: only if it affects the mitral valves (line 38).

Do have any control group? E.g. cardiac surgery patients without sepsis.

What statistics did you perform in Figure 1.

You should only use one level of significance say < 0.05. Seek statistical advice. (eg. Line 126).

What is the genetics and biochemistry of alkaline phosphatase (line 172). Which isoforms do you mean? Does the alkaline phosphatase in your study origin in kidney or liver or bone or is that unknown? If such data in patients are lacking what about animal data. LPS in mice is an accepted model for sepsis. Please discuss the biochemistry and genetics (KO mice for toll like receptor) of LPS in mice and alkaline phosphatase: see Singh SB, Lin HC. Role of Intestinal Alkaline Phosphatase in Innate Immunity. Biomolecules. 2021 Nov 29;11(12):1784. doi: 10.3390/biom11121784. PMID: 34944428; PMCID: PMC8698947.

The Discussion stops too sudden. What is your Summary and outlook for further work?

Reviewer 2 Report

This manuscript study by Amila Kahrovic et al. Present a “What about alkaline phosphatase in endocarditis?”. The title looks nun-scientific, but the manuscript's well-organized story and presented long-term data.

Authors show that Alkaline Phosphatase levels are important for patient survival. 

It is wondering based on the method section statistical analysis mentions the use of mean and standard deviation (SD) but Figure 1 uses 95% CI. Is it any reason for it?

In Figure 2: Is the baseline are same between a and b? if same, how about merging a and b like baseline, AP POD1, and AP POD30 with No and Yes? Also, what is the * symbol over each bar graph? Is it a different data point than the circle symbol?

In Figure 3: is it the overall survival rate? And what is the data present below each bar graph label? Is the survival patient number?

N/A

Round 2

Reviewer 1 Report

All my concerns have been met.